# Caught in the Web—Emotional Regulation Difficulties and Internet Addiction Among Romanian Medical and Technical University Students: A Pilot Cross-Sectional Study

**DOI:** 10.3390/healthcare13192528

**Published:** 2025-10-06

**Authors:** Simona Magdalena Hainagiu, Simona Nicoleta Neagu

**Affiliations:** The Department of Teacher Training and Social Sciences, National University of Science and Technology POLITEHNICA Bucharest, 060042 Bucharest, Romania; simona.hainagiu@upb.ro

**Keywords:** internet addiction, emotional regulation difficulties, medical students, technical students, educational stress, emotional education, emotional regulation support, mental health

## Abstract

Background and Objectives: Young people of all ages are now prematurely overexposed to a tech-addicted life, with negative psychological, physiological, sociological, and educational effects. Ease of access to and normalization of exposure to technology are indicated as the main causes of internet addiction and a mental health concern, especially in Romania, a country with widespread and easy access to the internet. Methods: This exploratory cross-sectional study with 132 participants investigated the relationship between emotional regulation difficulties (ERDs) and the risk of internet addiction (IA) among medical and technical students—two educational cultures defined by intense educational and emotional stress—with the aim of identifying specific patterns of variability. Standardized self-report data were collected, and specific descriptive and correlational statistical methods were used. Results: Key findings suggest similar moderate difficulty in emotional regulation for each student sample and normal-to-mild internet use for technical and medical students. A moderately strong Pearson correlation was observed between internet addiction and emotional regulation difficulties across the entire group of students (r = 0.54, *p* < 0.001). However, higher levels of emotional dysregulation and internet addiction risk were evident for the medical students. Conclusions: These results suggest that IA is closely linked to ERD rather than to the exposure to technology itself, as we presumed in the case of technical students. Moreover, medical students have a greater need for institutional support measures than their technical peers to cope with a highly challenging educational environment that exceeds individual levels of effective self-regulation.

## 1. Introduction

With the proliferation of smartphones, social media, and instant access to the internet, digital technologies have become deeply embedded in everyday routines. Problematic or excessive internet use, commonly referred to as Internet Addiction (IA), has emerged as a public health concern due to its psychological, social, and educational consequences. Emotional Regulation Difficulties (ERDs) are increasingly recognized as a central vulnerability, since impaired emotion regulation is both a risk factor for and a potential outcome of IA. Adolescents spend more time using media than they do sleeping or in school—an average of 7 h 22 min a day [1,2]. This excessive use can lead to loss of control, neglect of responsibilities, and social withdrawal, features consistent with behavioral addictions. A meta-analysis of 53 studies across 17 countries reported the global prevalence of IA as 3.05% [3], while adolescents and young adults showed an even higher vulnerability, with a prevalence of 9.9% for internet gaming disorder [4]. Beyond prevalence, IA has been linked to anxiety, depression, sleep problems, and academic impairment [5,6,7]. A recent meta-analysis [8] found strong associations between IA and measures of emotional dysregulation, particularly impulsivity and alexithymia. Furthermore, longitudinal studies suggest that ERD may predict the development of IA symptoms over time [9].

Romania has seen a rapid increase in internet accessibility and smartphone use, particularly among students. According to the National Institute of Statistics in 2024, 88.6% of households had internet access, with 98.9% among households with dependent children [10]. While this expansion brings educational benefits, it also raises risks of IA as online behaviors may easily shift from utility to IA-related risks [11] as it was revealed by the COVID-19 pandemic experience. In Romania, research on internet addiction is still scarce. Existing Romanian studies have documented high internet use among adolescents and young adults, with associations to emotional distress, depression, poor academic performance, impaired social functioning, and physical symptoms [12,13,14,15,16]. There is a real need for national studies that highlight protective factors in regulating addictive internet behavior, such as intelligence level or age. However, comprehensive national-level data remain scarce, and no study has directly compared medical and technical university students.

The term IA was originally introduced by Young as “an impulse-control disorder that does not involve an intoxicant” [17]. Although dysfunctional or addictive internet use is not formally recognized in DSM-5 or ICD-10 (except for Internet Gaming Disorder) [18], numerous studies have emphasized its negative biopsychosocial consequences, including structural brain changes, comorbid psychiatric symptoms, and increased risk of substance use [19,20,21]. At the same time, ERD has been consistently linked to a wide range of addictive behaviors. Meta-analyses and systematic reviews show that impulsivity, poor emotion regulation strategies, and lack of emotional awareness are common across both substance-related and behavioral addictions [22,23,24,25]. Effective emotional regulation is therefore a protective factor for mental health [26], while ERD has been associated with diverse psychopathological manifestations [27,28,29]. Effective emotional regulation is therefore a protective factor for mental health [26], while ERD has been associated with diverse psychopathological manifestations [27,28,29].

Taken together, these findings highlight the importance of examining how different educational environments may shape the link between IA and ERD. Medical and technical students represent an especially relevant population for such an investigation. Both groups face highly demanding academic contexts, but their stressors differ significantly. Technical students are frequently exposed to prolonged computer and internet use for academic tasks such as programming, simulations, or online resources, which may blur the boundary between functional and problematic use. In contrast, medical students experience intense psychological and emotional strain due to long study hours, frequent exams, and preparation for emotionally challenging clinical work. These contrasting contexts provide a unique opportunity to examine whether vulnerability to IA is primarily driven by technology exposure (as often presumed for technical students) or by emotional regulation difficulties under high academic stress (as hypothesized for medical students). To date, no national-level study has directly compared these two educational groups in Romania. Addressing this gap can provide insights into how educational environments shape risk factors for IA and guide institutional support programs. Therefore a Romanian-specific study is not only academically relevant but also socially and practically valuable.

The primary objective of this pilot cross-sectional study was to examine the relationship between IA and ERD among Romanian university students. Specifically, the study aimed (1) to assess and compare the levels of IA and ERD in medical versus technical students, (2) to test the association between IA and ERD across the total student sample, and (3) to explore whether medical students exhibit higher vulnerability to IA due to greater emotional regulation difficulties compared to their technical peers.

## 2. Materials and Methods

### 2.1. Study Design

This study employed a pilot cross-sectional design, chosen to explore the relationship between Internet Addiction (IA) and Emotional Regulation Difficulties (ERDs) in a small convenience sample of Romanian university students. The “pilot” designation reflects both the limited sample size and the exploratory aim of generating preliminary data to inform future larger-scale and representative studies. The cross-sectional nature of the design allowed the simultaneous collection of information on IA and ERD at a single point in time, which is appropriate for examining initial associations without implying causality.

### 2.2. Participants and Procedure

The study sample was selected using a snowball convenience sampling strategy, recruiting participants from medical and technical universities in Romania. The sample size was determined pragmatically, reflecting the exploratory and resource-limited nature of a pilot study. No formal power calculation was conducted, as the primary aim was to generate preliminary data and assess feasibility for future research. Accordingly, the findings provide indicative rather than representative insights into the broader population of Romanian medical and technical students.

Students were invited to participate voluntarily and anonymously, and the survey was disseminated online. The online survey was administered between April and May 2025, prior to the beginning of the examination period, in order to minimize the confounding effect of acute academic stress. Participation was voluntary and fully anonymous. To prevent multiple submissions from the same participant, the survey platform restricted responses to a single entry per device. Participants were asked to complete a set of questionnaires that assessed internet use and emotional regulation, along with sociodemographic characteristics. The survey link was distributed across various media (institutional email addresses, e-learning platforms, Facebook pages of student societies), displaying posters with the QR code in areas frequented by students (hallways, libraries, classroom entrances) across the student community from each university. The inclusion criteria were age ≥ 18 years, active enrollment in a university program (medical or technical), and informed consent for participation. Exclusion criteria were incomplete responses or refusal to consent. No additional exclusion criteria regarding psychiatric disorders or substance use were applied. Eligibility was assessed through an opening consent form and verification of university enrollment status before proceeding with the survey.

### 2.3. Instruments

Internet Addiction was assessed using Young’s Internet Addiction Test (IAT), a 20-item self-report questionnaire rated on a 5-point Likert scale, widely validated for identifying problematic internet use. The instrument has been applied across diverse populations and shows good reliability and validity, including in European and Romanian samples [17,30].

Emotional Regulation Difficulties were measured with the 36-item Difficulties in Emotion Regulation Scale (DERS), which evaluates multiple facets of emotion regulation across six subscales (Non-Acceptance, Goals, Impulse, Awareness, Strategies, Clarity) and yields a total score (range 36–180), with higher scores indicating greater difficulties. The original 36-item DERS has demonstrated robust construct validity and internal consistency, and the scale has been validated in Romanian samples. The DERS is a measure of regulation competence rather than a diagnostic instrument and does not provide clinical cut-offs [31,32,33,34,35].

### 2.4. Statistical Analysis

Data analysis was performed using Microsoft Excel (MS Office 365 package). Before data analysis, responses provided by volunteers were manually screened by the authors of the present study to identify potential bot interference or incomplete answers. Descriptive statistics (means, standard deviations, frequencies, and percentages) were calculated for main variables. Group comparisons between medical and technical students were conducted using independent samples t-tests for continuous variables and chi-square tests for categorical variables. Pearson correlation coefficients were used to assess associations between Internet Addiction (IA) and Emotional Regulation Difficulties (ERD). Statistical assumptions for parametric tests (normality and homogeneity of variances) were checked prior to analysis. The F-test for homogeneity of variances indicated no significant difference between the DERS and IAT scores (F = 1.09, *p* = 0.32), DERS CDUMPB and DERS NUSTPB (F = 1.08, *p* = 0.37) and IAT CDUMPB and IAT NUSTPB (F = 1.38, *p* = 0.10), suggesting that the variances of the two groups are comparable. Histograms and cumulative frequency plots indicate that most participants had low to moderate scores, with a right-skewed distribution and few high-score outliers. This suggests that the assumption of strict normality may be mildly violated, but the distribution is approximately symmetric for parametric analyses. The distributions of DERS and IAT scores were assessed for normality using skewness and kurtosis. The values were within commonly accepted limits (|skewness| < 2, |kurtosis| < 7), indicating that the distributions can be considered approximately normal. Significance levels were set at *p* < 0.05, in accordance with standard practice in the field. Sociodemographic data such as gender distribution, academic year, habitation were collected and are presented in the Results Section.

### 2.5. Ethical Considerations

In line with the current ethical standard research procedure, informed consent was collected from all participants before completing our battery of instruments, specifying all necessary information according to the law, including the fact that they could withdraw at any time without any consequences. Therefore, the current research follows the principles of the 2013 Helsinki Declaration and was approved by The Ethics Committee of the Department to which the authors are affiliated (Protocol Number 117/08.04.2025). Participants were informed at the beginning of the online survey about the study’s aims, procedures, and their rights, including the voluntary nature of participation and the possibility to withdraw at any time without consequences. Informed consent was obtained electronically by requiring participants to read the information sheet and actively select an agreement option before accessing the questionnaire. No identifying information (such as names, email addresses, or IP data) was collected. Responses were stored securely on a protected server accessible only to the research team, ensuring compliance with data protection standards and confidentiality throughout the study.

## 3. Results

The study involved 132 university students from National University of Science and Technology POLITEHNICA Bucharest (NUSTPB) (90 students) and Carol Davila University of Medicine and Pharmacy in Bucharest (CDUMPB) (42 students), Romania, enrolled in undergraduate and postgraduate studies, aged between 19 and 26 years old. Out of the total number of participants, 84 (63.64%) are female, while 47 (35.61%) are male, and 1 is nonbinary (0.76%). Regarding the level of education 75.76% students are enrolled in undergraduate programs, while 24.24% participants are enrolled in postgraduate programs in both universities. The distribution by university according to gender is presented in Table 1.

Other sociodemographic information about participants can be found in Table 2 (academic program type) and Table 3 (living conditions). For each university, we differentiated between the undergraduate and postgraduate programs, as at the CDUMPB, the bachelor’s program lasts for six years, while at NUSTPB, the bachelor’s program lasts for four years.

The DERS Scores were analyzed for the entire group of participants, taking into account the total score, as well as for each sample of subjects according to university and gender, as shown in Figure 1.

The response values were analyzed according to the specific categories for the 36 items of DERS. The values obtained (M = 95.69, SD = 23.61 scores for the CDUMPB students and M = 91.9, SD = 22.66 for the NUSTPB students) show a moderate level of difficulty in emotional regulation both at the general level of the student group and for each university: students from CDUMPB and NUSTPB recorded similar average values on the DERS scale. The men from CDUMPB have more serious difficulties in emotional regulation than the women from the same university, but also higher scores than the men from NUSTPB, which are even lower than their female colleagues. The women from NUSTPB have more difficulties in regulating their emotions than the men from the same university. The women from NUSTPB have also higher scores on DERS than the women from CDUMPB. However, men at CDUMPB have the greatest levels of emotional difficulties among all students.

For a more in-depth analysis of the DERS scores obtained by the two student samples, Cohen’s score was calculated, as shown in Table 4.

A comparison between the CDUMPB (M = 95.69, SD = 23.62, *n* = 42) and NUSTPB (M = 91.90, SD = 22.66, *n* = 90) groups suggests a statistically insignificant difference between means (t State = 0.87, t Critical one-tail = 1.99, *p* = 0.39), with a small effect size (Cohen’s *d* = 0.17) as it is presented in Table 4.

Regarding the IAT results, the total score for the entire group of subjects was calculated, as well as the scores for each sample of students, depending on the university and gender, as shown in Figure 2.

The total score for the entire group of participants (30 points) corresponds to a Normal Use category describing an average internet user, occasionally going online longer than intended, but maintaining control [17]. Analyzing the student sample from each university, we can see that students in technical programs maintain normal internet usage (M = 28.21, SD = 14.46), while those in medical programs show a mild dependence on the internet (Mean = 32.26, SD = 17.00). The highest level of internet addiction is among male medical students (34 points), while the lowest level of addiction is among female technical students (28 points).

For a more in-depth analysis of the IAT scores obtained by the two student samples, Cohen’s score was calculated, as shown in Table 5.

A comparison between the CDUMPB (M = 32.26, SD = 17.01, *n* = 42) and NUSTPB (M = 28.21, SD = 14.47, *n* = 90) groups revealed a non-significant difference (t State = 1.33, t Critical two-tail = 1.99, *p* = 0.19), with a small effect size (*d* = 0.26). Although the difference did not reach statistical significance, the effect size suggests a meaningful trend, indicating that medical students tend to show slightly higher levels of internet addiction compared to technical students. In the context of mental health research, even small differences such as this may have practical relevance.

Based on this partial analysis for each group of students, for a more comprehensive view of the two dimensions investigated, we conducted a correlational analysis across the entire sample of participants. As shown in Table 6, Pearson’s correlation analysis (r = 0.54, *n* = 132, *p* < 0.001) revealed a statistically significant moderate positive relationship, as shown in Table 6, using the r magnitude classification (0.1–0.3 = weak correlation, 0.3–0.5 = moderate correlation, 0.5–0.7 = moderately strong correlation, 0.7+ = strong correlation).

The results show that greater difficulties in emotion regulation are associated with higher levels of problematic internet use in this population. While causality cannot be inferred, the strength of the association warrants further investigation into targeted emotional regulation interventions.

## 4. Discussion

Contrary to our initial expectation regarding the higher risk of emotional dysregulation and internet addiction among students enrolled in technical programs, the data analysis reveals surprising results: the finding that medical students in this sample showed a non-significant trend toward higher internet addiction levels than their technical peers points to the role of academic and emotional strain in predisposing them to possible internet addictive behavior. However, given the small effect size and lack of statistical significance, this observation should be interpreted with caution and considered as exploratory, warranting confirmation in larger and more representative samples. Rather than mere exposure to technology, it appears that academic and emotional strain could play a role in addictive internet use, a pattern consistent with prior work linking stress and emotional overload to behavioral addictions [12,22,25], as well as a hypothesis for future research. Our results could be relevant in the context of the previous data showing that gender has a small or no impact on DERS scores [36]. Our exploratory findings suggest that medical students in this sample may show a higher tendency toward problematic internet use, potentially as a coping mechanism in response to academic or personal stressors. While internet addiction has been associated with adverse outcomes such as poor academic performance, anxiety, and reduced emotional control [17,37], our results suggest that certain student populations may be less susceptible to these risks.

Given the pilot and non-representative design, these results should be interpreted with caution and not generalized to all medical student populations. Supporting this, the greater ERD observed in medical students, especially among men, suggests that difficulties in managing emotions may exacerbate reliance on maladaptive coping strategies such as excessive internet use. This gender-specific pattern adds nuance to existing findings, which often emphasize female vulnerability, and underscores the importance of addressing emotion regulation skills across both genders in preventive interventions [27,28,29]. The co-occurrence of elevated IA and ERD in the medical student subgroup reinforces the theoretical assumption that impaired emotion regulation constitutes a central vulnerability pathway to behavioral addictions [8,9,23]. Specifically, difficulties in managing emotions may contribute to maladaptive behaviors such as excessive internet use. The pattern observed in male medical students from our sample points to the potential importance of considering gender differences in preventive interventions. However, due to the limited and uneven sample, further studies are needed before firm conclusions can be drawn. Together, these findings highlight the need for tailored mental health support and coping strategies that address both emotional regulation and internet use behaviors, particularly within the demanding context of medical education. Further research is warranted to explore the causal relationships and to develop targeted programs aimed at improving emotional resilience and reducing the risk of IA among medical students.

It is also relevant that IA and ERD levels in our sample fell within the moderate-to-normal range, which may suggest that higher education could exert a protective effect by fostering adaptive coping skills and cognitive control. As online behaviors shift from utility to dependency or danger [11], especially during and after the COVID-19 pandemic, it is crucial to understand how these changes are affecting mental health and emotional regulation of young individuals who are still in the stage of developing cognitive and emotional regulation mechanisms, even if they are considered legally autonomous and responsible adults. This interpretation resonates with prior studies reporting that demanding academic environments, despite their stress, may cultivate resilience in some student populations [38,39]. In the context of previous research that has revealed higher levels of IA at younger ages, we can encourage an optimistic outlook on the evolution of IA over time, as well as consider the protective factor of higher education, which moderates the level of IA and supports the development of emotional skills. 

An important consideration is that the observed associations may also be shaped by alternative explanations and potential confounding factors. For instance, differences in the purpose of internet use could partly explain the results: technical students are likely to spend considerable time online for study-related activities (e.g., programming, simulations), whereas medical students may use the internet more frequently for social or coping purposes, which could inflate IA scores without reflecting a purely addictive pattern. In addition, the snowball convenience sampling method may have introduced self-selection bias, favoring participation from students who are more active online or more interested in the study’s theme, thereby limiting representativeness. Moreover, while our findings show a moderate association between ERD and IA, the cross-sectional design prevents any conclusions about directionality. It is plausible that ERD contributes to problematic internet use, but equally possible that excessive internet use exacerbates emotional regulation difficulties, or that both share common underlying vulnerabilities such as impulsivity or anxiety. Future longitudinal and experimental studies are necessary to disentangle these complex relationships.

### Limitations

This study has several limitations that should be acknowledged. First, the use of snowball convenience sampling facilitated access to participants in a cost-effective and timely manner, but introduces selection bias, limiting the generalizability of the findings, as the sample may not accurately represent the broader population of Romanian medical and technical students. This limitation is acknowledged and further addressed in the Discussion section. Not excluding participants with psychiatric or substance abuse history reflects the exploratory and feasibility-oriented nature of the pilot study, which aimed to capture a broad snapshot of student populations without introducing further barriers to participation. However, we acknowledge that psychiatric comorbidities and substance use may influence IA and ERD, and the absence of such screening represents a limitation to be addressed in future research. Second, the sample size was modest and uneven across groups, further reducing representativeness and limiting the statistical power to detect differences. Third, data were collected through self-report questionnaires, which are vulnerable to response biases such as social desirability and inaccurate self-assessment. Fourth, the cross-sectional design precludes conclusions about causality or directionality; while ERD and IA were associated, the temporal order of these variables cannot be established, and reverse or bidirectional effects remain plausible.

This non-probabilistic approach allowed rapid access to respondents at low cost, but it also introduced selection bias. Specifically, participants were recruited through peer networks, meaning that individuals with similar characteristics (e.g., study habits, internet use behaviors, or social backgrounds) were more likely to be included. As a result, the sample may over-represent students with higher online engagement or stronger social connections, while under-representing those less integrated into these networks. Future research should employ larger, randomized, and more diverse samples to enhance external validity.

Another limitation of the study is the exclusive reliance on self-report instruments for assessing internet use and emotional regulation difficulties. While self-reports are widely used and validated, they may be subject to recall bias and social desirability. Future studies should aim to triangulate these measures with objective data, such as app tracking or system logs, to provide a more accurate and comprehensive assessment of internet use patterns. Due to the limited sample size, we were not able to conduct advanced statistical analyses such as mediation or moderation models. Such approaches (for example, testing whether gender moderates the association between ERD and IA) would provide more detailed insights into mechanisms and subgroup vulnerabilities. Future studies with larger and more representative samples should incorporate such analyses. Furthermore, our study lacks the robustness and advanced statistical capabilities of dedicated software that we could consider in future studies, while Excel allowed basic descriptive, correlational and t-test analyses in the present study.

Both groups of students encounter demanding workloads, performance pressure, and prolonged exposure to digital environments, which may contribute to emotional exhaustion [40]. In such contexts, the internet can act as a readily available tool for temporary emotional relief, yet this reliance may foster maladaptive coping and reinforce addiction-like patterns [41]. Just as the integration of artificial intelligence in education requires careful regulation [42], the use of digital technologies by students calls for effective self-regulation mechanisms. Internet use behaviors may have been shaped by the residual effects of the COVID-19 pandemic, which significantly altered patterns of digital engagement and coping strategies among young people worldwide. Moreover, cultural differences may also play a role in shaping both emotional regulation and internet use behaviors; the patterns observed in this Romanian student population may not fully correspond to those in other national or cultural contexts. Another limitation is the absence of control scales for depression, anxiety, burnout, and social support. These factors are known to be associated with both internet use behaviors and emotional regulation, and their omission prevents a clear assessment of the unique contribution of ERD to IA. Future research should include such measures.

Understanding the interplay between Internet Addiction (IA) and Emotional Regulation Difficulties (ERDs) is essential for developing effective prevention and intervention strategies. Our findings support previous evidence that emotion regulation plays a central role in problematic internet use. Enhancing students’ ability to regulate emotions may therefore represent a protective factor, helping to reduce the risk and severity of internet-related problems. Given the increasing digitalization of human interaction, such interventions are especially relevant for young populations, who are still consolidating their cognitive and emotional coping mechanisms.

## 5. Conclusions

This pilot cross-sectional study examined the relationship between Internet Addiction (IA) and Emotional Regulation Difficulties (ERDs) among Romanian medical and technical students, highlighting the influence of academic and emotional stressors. While these findings provide valuable insights, they must be interpreted with caution due to the study’s exploratory nature, small and non-representative sample, and reliance on self-report measures. Therefore, the results cannot be generalized to all Romanian medical and technical students.

Future research should use larger, representative samples, longitudinal designs to assess causality, and intervention trials to evaluate programs aimed at strengthening coping and emotional regulation skills. From a practical perspective, higher education institutions—particularly medical faculties—should consider integrating preventive programs that address sleep hygiene, stress management, and emotional regulation training. From a practical perspective, these findings highlight the need for context-aware preventive measures in the Romanian universities we involved in the study. Medical faculties, where students appear particularly vulnerable, could integrate training in emotional regulation and adaptive coping strategies into the curriculum. Regular mental health screenings and counseling services could help identify students at risk early, while awareness campaigns on healthy internet use may prevent maladaptive patterns from becoming entrenched. For technical students, interventions may focus more on balancing instrumental technology use with digital well-being practices. Gender-sensitive approaches, including targeted support programs for female students, are also warranted. Collectively, these measures could help institutions foster both academic success and psychological resilience.

In conclusion, this pilot study emphasizes the importance of addressing both emotional regulation and internet use behaviors in student populations, offering a foundation for targeted future research and institutional strategies to promote academic resilience and mental health.

## Figures and Tables

**Figure 1 healthcare-13-02528-f001:**
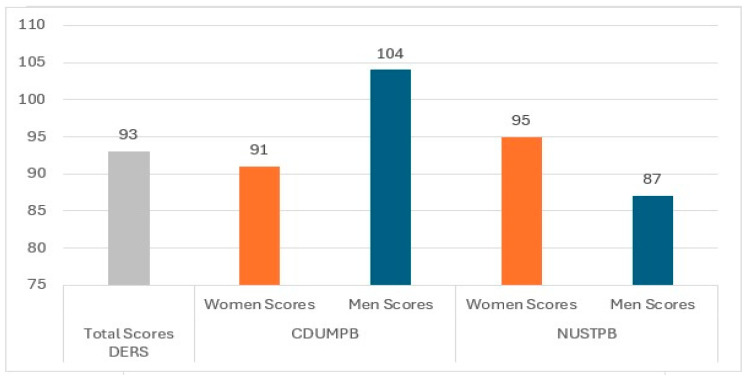
The DERS score distribution by university and gender.

**Figure 2 healthcare-13-02528-f002:**
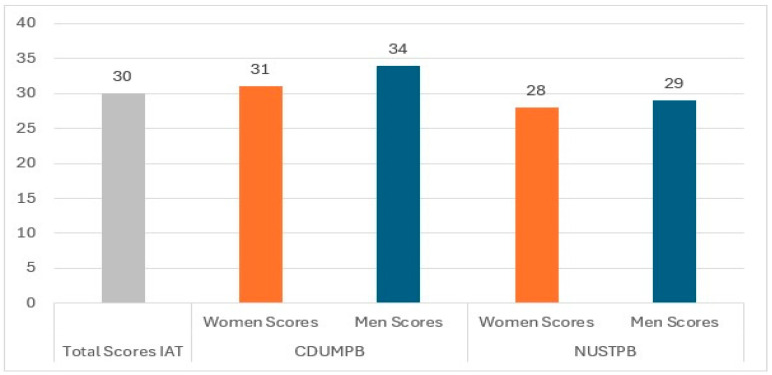
The IAT score distribution by university and gender.

**Table 1 healthcare-13-02528-t001:** Participant distribution by university and gender.

University	Women	Men	Nonbinary	Total
CDUMPB	30	11	1	42
NUSTPB	54	36		90
Total	84	47		132

**Table 2 healthcare-13-02528-t002:** Participant distribution by university and academic program type.

**CDUMPB**	Undergraduate	42 (32%)
**NUSTPB**	Undergraduate	58 (45%)
	Postgraduate	32 (23%)
	Total	132 (100%)

**Table 3 healthcare-13-02528-t003:** Participant distribution by university and living conditions.

**CDUMPB**	Living with roommates	12 (9%)
	Living with family	13 (10%)
	Living alone	17 (13%)
**NUSTPB**	Living with roommates	32 (24%)
	Living with family	35 (27%)
	Living alone	23 (17%)
	Total	132 (100%)

**Table 4 healthcare-13-02528-t004:** Cohen’s effect size of the difference between the means of the two groups of students on the DERS scale.

	MEAN	SD	*n*
CDUMPB	95.69	23.62	42
NUSTPB	91.90	22.66	90
M1-M2	3.79		
SD	22.97		
Cohen’s *d*	0.17		

**Table 5 healthcare-13-02528-t005:** Cohen’s effect size of the difference between the means of the two groups of students on the IAT.

	MEAN	SD	*n*
CDUMPB	32.26	17.01	42
NUSTPB	28.21	14.47	90
M1-M2	4.05		
SD	15.31		
Cohen’s *d*	0.26		

**Table 6 healthcare-13-02528-t006:** DERS-IAT correlation among the entire group of students.

	IAT	DERS
**IAT**	1	
**DERS**	0.54 *	1

* Pearson’s correlation is significant at the 0.01 level (2-tailed), *p* = 0.000.

## Data Availability

The raw data supporting the conclusions of this article will be made available by the authors on request.

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
