# Peer review of "Caught in the Web—Emotional Regulation Difficulties and Internet Addiction Among Romanian Medical and Technical University Students: A Pilot Cross-Sectional Study"

_healthcare, 2025, doi:10.3390/healthcare13192528_

Round 1

Reviewer 1 Report

Comments and Suggestions for Authors

The present pilot study explores the relationship between emotional regulation difficulties (DERS) and the risk of internet addiction (IAT) in a sample of 132 Romanian university students (42 medical students and 90 engineering students). The findings indicate a moderate positive correlation (r = 0.54) between the two dimensions. Notably, medical students exhibited higher levels of both emotional dysfunction and IA compared to engineers. The authors conclude that vulnerability to IA is more related to emotional regulation difficulties than to mere technological exposure, emphasising the need for targeted interventions for future doctors. The topic is of great interest, and it is important to emphasise several positive aspects. Firstly, the topic is certainly timely and socially relevant, with a specific focus that has been little studied until now. The researchers must also be commended for their utilisation of standardised and validated tools (IAT and DERS) in the context of the local population. Furthermore, the research methodology is founded upon a meticulously delineated cross-sectional design, accompanied by a discernible recruitment procedure and adherence to ethical standards. This has enabled a more nuanced analysis by subgroups (university and gender), accompanied by Cohen's effects, which facilitate assessments of practical significance. However, the work is not without its weaknesses. Primarily, the sample is limited in size (n = 132) and imbalanced (3:2 for engineers vs. doctors), resulting in diminished statistical power and restricted generalizability. Nevertheless, this predicament emanates from the selection of the investigative subject. The cross-sectional design of the study is a potential area for improvement, as it does not currently permit causal inferences to be made between emotional dysfunction and AI. Additionally, the absence of control for relevant covariates, including age, year of study, workload, hours of online study, and social support, should be noted. The absence of objective behavioural data, such as connection logs or actual time of use, that would provide substantiation for the self-assessment is a notable concern. This could result in the occurrence of selection bias (snowballing via social media) and the potential presence of socially desirable responses. It is evident that the text could be enhanced in the following ways for the reasons outlined below:

  1. In order to test the directionality of the relationships, it is necessary to integrate a longitudinal design (baseline and follow-up) or at least repeated measures.
  2. The objective data on internet use, such as app tracking and system logs, should be collected to triangulate the self-reports.
  3. 3. It is recommended that mediation/moderation analysis be incorporated (e.g., gender as a moderator) in order to elucidate mechanisms and subgroups at risk.The "Limitations" section should be expanded in a transparent manner, with discussion of the possible influence of sampling, the COVD-19 pandemic, and cultural differences.Finally, it is suggested that control scales for depression, anxiety, burnout, and social support be included to assess the unique effect of emotional regulation. It is acknowledged that this would significantly delay the publication of the paper.
  4. However, as previously stated, this is an additional point that does not necessarily have to be adhered to.

The length of this review is regrettable, but the research is of significant value and these aspects would render it even more compelling.

Reviewer 2 Report

Comments and Suggestions for Authors

Dear Author(s),

Thank you for submitting your manuscript entitled “Caught in the web – Emotional Regulation Difficulties and Internet Addiction among Romanian Medical and Technical University Students: A Pilot Cross-Sectional Study” to Healthcare.

This pilot cross-sectional study examines the association between emotional regulation difficulties (ERD) and internet addiction (IA) among Romanian university students in two distinct educational contexts: medical and technical programs. The topic is interesting, also considering the global increase in problematic internet use and its psychological correlates. The focus on Romania may fill an important gap in the literature, where national-level data remain scarce. The study benefits from the use of validated instruments and a combination of descriptive and correlational analyses, offering potentially valuable insights for mental health promotion and educational strategies in higher education.

Overall, the manuscript is supported by an extensive background, and the study has merit in its originality and contextual relevance. However, several aspects require clarification and deeper critical reflection to enhance the paper’s clarity, methodological transparency, and overall impact.

Below are my comments and suggestions, organized by section.

INTRODUCTION

The introduction provides an extensive review of the literature on IA and ERD, encompassing both global and Romanian contexts, and establishes the potential relevance of the study. However, it is overly long, inconsistent in style, occasionally repetitive, and sometimes blurs the distinction between background and interpretation. The research gap is mentioned but not sharply defined.

Main issues:

  • Length and redundancy: Several ideas and statistics are repeated (e.g., multiple mentions of the scarcity of Romanian studies and reiteration of prevalence figures).
  • Unclear research gap: The need to focus specifically on medical/technical students is not fully justified.
  • Overlap with discussion: Some content (e.g., the paragraph starting “Understanding the interplay between IA and ERD can inform the…”) would be better placed in the Discussion. The introduction should lay to background and rationale, leaving implications to the discussion.
  • Excessive statistical detail: Several studies are described in excessive detail (sample sizes, daily usage hours, percentages) without always linking these directly to the current study’s purpose. Select only the most relevant figures to support the argument.
  • Language and style: Minor grammar and spelling errors are present (e.g., “tehnical” instead of “technical”; “copyig” instead of “coping”). IA and ERD are introduced late in the paragraph, though they appear early; ensure abbreviations are introduced upon first mention.
  • Flow: Transitions between global context, Romanian data, and the theoretical framework could be smoother.

Recommendations:

  • Condense and focus the literature review, integrating similar points and removing repetition.
  • Clearly articulate the specific knowledge gap and the novelty of comparing medical and technical students.
  • End the section with a concise, explicit statement of study objectives.
  • Proofread for grammar and ensure consistent abbreviation use.
  • Reorganize for logical flow: Suggested sequence:
    • Definition and global significance of IA and ERD.
    • Global prevalence and consequences.
    • Romanian context and data gaps.
    • Link between ERD and IA (global evidence).
    • Rationale for focusing on medical/technical students.
    • Study aim and hypothesis.

METHODS:

The Methods section contains key details on instruments, participants, and procedures, but requires improvements in structure, transparency, and completeness to ensure reproducibility.

Main issues:

  • Study design: The study design (“pilot cross-sectional” as reported in the Abstract) should be stated in the opening sentence to ensure clarity, and justified (e.g., small sample, exploratory aim…).
  • Sampling and limitations: Sampling via snowball convenience sampling introduces selection bias; this must be acknowledged in a Limitations section (currently absent in the manuscript – please add) and considered when discussing the results of the study (please adjust accordingly).
  • Sample size and rationale: No rationale is given for the sample size, even in the context of a pilot study. The representativeness of the sample for the broader population of Romanian medical and technical students is unclear.
  • Eligibility criteria and screening: Eligibility criteria are unclear, particularly whether individuals with psychiatric disorders or substance use were included/excluded, considering the potential impact of these conditions on IA and ERD. Furthermore, it is unclear how eligibility was assessed (e.g., did the survey contain an opening form to assess eligibility?).
  • Missing procedural details: survey duration, anonymity measures, and prevention of multiple submissions are not reported, undermining the transparency and scientific rigour of the study.
  • No dedicated statistical analysis subsection: a subsection on tests used, assumptions checked, significance levels, and software employed is not reported, while this information should be specified to ensure clarity and reproducibility.
  • Misplacement of results: Some results appear prematurely in this section that should be focused on methods (e.g., participants' distribution by university and gender). These data should be reported in the Results section. However, sociodemographic data (mean age, marital status, academic year, etc.) are not reported. Please add this data or, if absent, consider this a limitation of the study.
  • Ethical consideration: Ethical approval is mentioned, but this section lacks detail on data protection, anonymization, and storage. Furthermore, it is unclear when and how participants were informed about the study's purposes and provided informed consent for participation.
  • Overlong instrument descriptions: While detail is good, the full interpretive categories for the IAT and DERS could be shortened to avoid excessive length, especially when these are already documented in the cited literature.

Recommendations:

  • Add a clear study design statement and rationale for the pilot cross-sectional study design.
  • Provide a sample size justification.
  • Detail eligibility criteria and consent procedures.
  • Create a dedicated statistical analysis subsection.
  • Include any results and full sociodemographic profile (if present) in the Results section.
  • Expand the ethics statement to meet international standards.

RESULTS:

The Results section contains extensive descriptive, comparative, and correlational data, supported by tables and figures. However, the narrative is overly descriptive and sometimes overlaps with interpretation.

Main issues:

  • Repetition of table/figure content in the text: please avoid redundancy.
  • Mixing interpretation with results: Several statements move beyond reporting data into speculation (e.g., “We can say that it is more challenging for women at NUSTPB to regulate their emotions…”). Interpretive comments should be reserved for the Discussion section.
  • Incomplete statistical reporting: p-values are missing for group comparisons, assumption testing is not mentioned, and effect size interpretation lacks thresholds.
  • Figures/tables are inconsistently labeled: some lack clear captions, axis labels, or context. Please consider that figures/tables should be self-explanatory.
  • Correlation analysis clarity: The Pearson correlation table is covered with unnecessary intermediate statistics (variance, df, t-critical values), which could be simplified to r, p-value, and N. The narrative claims “moderate correlation” without referencing the threshold used for classification.
  • Inconsistent terminology: The terms “addiction,” “dependence,” and “problematic use” are used interchangeably, which can confuse interpretation.

Recommendations:

  • Summarize key patterns rather than restating all numerical details.
  • Separate interpretation from results.
  • Report test statistics and p-values for all group comparisons.
  • Ensure all figures/tables are fully labeled and self-explanatory.
  • Use consistent terminology throughout.

DISCUSSION:

The Discussion links findings to existing literature and offers plausible explanations for the higher IA and ERD among medical students. Nonetheless, it suffers from repetition, limited critical appraisal, and overgeneralization.

Main issues:

  • Repetition of results: Several paragraphs restate descriptive findings almost verbatim from the Results section (e.g., “medical students have higher levels of IA compared to their peers in technical programs”). This reduces the interpretation and undermines the focus of this section.
  • Limited exploration of alternative explanations or confounding factors: The discussion focuses heavily on plausible explanations without addressing alternative interpretations or potential confounding factors (e.g., differences in internet use for study purposes, self-selection bias from the sampling method). The link between ERD and IA is taken at face value without discussing the possibility of reverse causality or shared underlying traits.
  • Absence of limitations subsection: The study limitations are not addressed (e.g., sampling bias, non-representative sample, self-report bias, cross-sectional design, uneven group sizes).
  • Overgeneralization of findings: The study is presented as if its results can be generalized to all Romanian medical and technical students, but the methods do not allow this. Statements such as “medical students may be more prone to problematic internet use” should be tempered to reflect the pilot and exploratory nature.

Recommendations:

  • Summarize results briefly and focus on interpretation.
  • Create a clear “Limitations” subsection.
  • Temper generalizations and emphasize the pilot nature.
  • Offer specific, context-aware policy or practice recommendations.

CONCLUSION

The Conclusion reiterates the main findings and practical implications but is overly long and repetitive, reading more like a continuation of the Discussion.

Main issues:

  • Repetition of discussion content: Several sentences rehash interpretations already presented earlier (e.g., academic demands, poor sleep, lack of coping skills in the curriculum). This reduces the impact of the conclusion.
  • Limited forward-looking perspective: While the text hints at future implications, it lacks explicit recommendations for future research (e.g., longitudinal studies, larger representative samples, intervention trials).
  • Overgeneralization: Statements such as “medical students face intense academic demands… and internet use may be a coping mechanism” are too general and read like literature review material, rather than study-specific conclusions. The findings should be qualified as exploratory, given the pilot and non-representative nature of the sample.

Recommendations:

  • Condense to core findings, key implications, and targeted recommendations for research/practice.
  • Acknowledge the study’s exploratory and limited nature.
  • End with a concise, impactful statement on the relevance of addressing ERD/IA in student populations.

Final consideration:

The manuscript addresses an important issue and uses validated tools to explore IA and ERD in a student population. With more concise writing, clearer articulation of the research gap, a stronger methodological description, more rigorous statistical reporting, and a sharper discussion and limitations, the study could make an interesting contribution to the literature.

I hope these comments are constructive and helpful to the Author(s).

Best regards.

Round 2

Reviewer 1 Report

Comments and Suggestions for Authors

The paper meets the requirements for publication following the revisions reported by the authors.

Author Response

We have reviewed the clarity of the presentation of the results, we have revised the Conclusions section in relation to the Results section, and we have optimized the presentation of the tables.

Reviewer 2 Report

Comments and Suggestions for Authors

Dear Author(s),

Thank you for submitting the revised version of your manuscript entitled “Caught in the web – Emotional Regulation Difficulties and Internet Addiction among Romanian Medical and Technical University Students: A Pilot Cross-Sectional Study” to Healthcare.

The manuscript has improved substantially since the previous version: the study design is now explicitly described, sociodemographic data are reported, ethical and procedural details are clarified, and a Limitations section has been added. The paper is more structured and readable, and the exploratory value of the study is now evident.

However, several serious statistical reporting problems remain, which directly affect the credibility of the findings. Unless corrected, these errors undermine the paper's main claims.
Below are my comments and suggestions.

  1. Correlation inconsistencies

The Abstract reports a Pearson correlation of r = 0.54 (page 1), labeled “moderate,” but without a p-value. The Results section reports r = 0.26, N = 132, p < 0.001, and also describes it as “moderate” (page 7). According to the manuscript’s own thresholds (reported on page 7), r = 0.26 is considered weak, not moderate. The correct coefficient, p-value, confidence interval, and magnitude label must be reported consistently across Abstract and Results.

  1. Group comparisons (IAT scores)

The comparison between medical and technical students is reported as t(130) = 1.45, p < 0.001, Cohen’s d = 0.26 (page 7). This is mathematically inconsistent: t = 1.45 with df = 130 corresponds to p ≈ 0.15 (that is non-significant). Either the t-value or p-value (or both) is wrong. Given the small effect size, the difference is likely not significant. The interpretation that medical students have significantly higher IA must be corrected or tempered to reflect a trend or suggestion if it is non-significant.

  1. Significance reporting

The Statistical Analysis section (page 4, section 2.4) states: “in our case the analysis revealed a value of p < 0.001.” This blanket statement is misleading, since not all analyses met this threshold. Each test must have its own exact p-value reported.

  1. Effect size interpretation

Small effects (e.g., d = 0.17, d = 0.26) are sometimes overstated as “meaningful” or “small to medium”. They should be explicitly described as small, and their exploratory interest acknowledged without exaggeration.

  1. Assumption checks

The manuscript notes that assumptions for parametric tests were checked (page 4, section 2.4), but no outcomes are reported. State whether normality and variance homogeneity were met, and how violations (if any) were addressed. If assumptions were not satisfied, non-parametric alternatives should have been considered.

  1. Transparency in methods

Data analysis was conducted in Excel; while acceptable for a pilot, this is a methodological limitation given the availability of more robust statistical packages.

  1. Abstract alignment

The Abstract currently describes “similar moderate risk of internet addiction” and a “moderate correlation”, which does not match the actual Results (predominantly normal/mild internet use, weak correlation). Revise the Abstract to reflect the true findings.

  1. Tables and figures

Table numbering is inconsistent (duplicate Table 2 on pages 5 & 6, and Table 3 on pages 5 & 7). Correct sequential numbering and ensure in-text references match.

  1. Language and formatting

Several minor errors remain: typos (“Disscussions,” “roomates,” “Cohens”), inconsistent decimal/p-value formatting, and template placeholders (“funding acquisition, Y.Y.”). These should be corrected in a final polish.

  1. Speculative interpretation in Discussion
  • On page 8, the statement “the psychological burden of medical training may be a more salient risk factor” is speculative. The study did not measure mental health outcomes or psychological burden, so such claims are unsupported.
  • Furthermore, the cross-sectional pilot design cannot identify risk factors, which require longitudinal data and regression modeling. If retained, this point must be reframed as a hypothesis for future research, not a conclusion of the present study.
  1. Methods vs. limitations

Section 2.2 (“Participants and Procedure”) contains justifications for methodological choices (e.g., not excluding participants with psychiatric history, snowball sampling). These are not methods but limitations and should be moved to the Limitations section. The Methods should describe what was done, not justify it.

  1. Limitations section content

The last part of the Limitations (page 10) discusses implications for practice, which are not limitations. This text should be renamed or relocated to the Discussion or Conclusion, leaving the Limitations section to focus strictly on methodological and interpretive constraints.

Conclusion
The manuscript has progressed significantly, but statistical errors and inconsistencies in reporting/interpretation remain major concerns. These issues must be corrected, as they directly affect the validity of the conclusions (particularly the claim of higher IA among medical students).

With corrected analyses, consistent reporting, and cautious interpretation, the study would be acceptable as a pilot, exploratory contribution to the literature on internet addiction and emotional regulation among Romanian students.

I hope these comments are constructive and helpful to the Author(s).

Best regards.
